# The RothC Model to Complement Life Cycle Analyses: A Case Study of an Italian Olive Grove

Valentina Fantin [1], Alessandro Buscaroli [2,3], Patrizia Buttol [1], Elisa Novelli [4], Cristian Soldati [3], Denis Zannoni [2,3], Giovanni Zucchi [4] and Serena Righi [3,5,*]

1 ENEA—Italian National Agency for New Technologies, Energy and Sustainable Economic Development, Laboratory for Valorisation of Resources in Productive and Territorial Systems (SSPT-USER-RISE), Via Martiri di Monte Sole 4, 40129 Bologna, Italy; valentina.fantin@enea.it (V.F.); patrizia.buttol@enea.it (P.B.)
2 Biological, Geological and Environmental Sciences Department (BiGeA), Campus di Ravenna, Alma Mater Studiorum Università di Bologna, Via S. Alberto, 16, 48123 Ravenna, Italy; alessandro.buscaroli@unibo.it (A.B.); denis.zannoni@unibo.it (D.Z.)
3 Inter-Departmental Centre for Research in Environmental Sciences (CIRSA), Alma Mater Studiorum—Università Bologna, Via S. Alberto 163, 48123 Ravenna, Italy; cristian.soldati@studio.unibo.it
4 Oleificio Zucchi Spa, Via Acquaviva 12, 26100 Cremona, Italy; evosustainability@oleificiozucchi.com (E.N.); vicepresident@oleificiozucchi.com (G.Z.)
5 Physics and Astronomy Department (DIFA), Alma Mater Studiorum Università di Bologna, Viale Carlo Berti Pichat 6/2, 40127 Bologna, Italy
* Correspondence: serena.righi2@unibo.it; Tel.: +39-0544-937306

**Abstract:** Soil organic carbon (SOC) plays a fundamental role in soil health, and its storage in soil is an important element to mitigate climate change. How to include this factor in Life Cycle Assessment studies has been the object of several papers and is still under discussion. SOC storage has been proposed as an additional environmental information in some applications of the Product Environmental Footprint (PEF). In the framework of wider activity aimed at producing the PEF of olive oil, the RothC model was applied to an olive cultivation located in Lazio region (Italy) to calculate the SOC storage and assess four scenarios representing different agricultural practices. RothC applicability, possible use of its results for improving product environmental performance, and relevance of SOC storage in terms of $CO_{2eq}$ compared to greenhouse gas emissions of the life-cycle of olive oil are discussed in this paper. According to the results, in all scenarios, the contribution in terms of $CO_{2eq}$ associated with SOC storage is remarkable compared to the total greenhouse gas emissions of the olive oil life-cycle. It is the opinion of the authors that the calculation of the SOC balance allows a more proper evaluation of the agricultural products contribution to climate change, and that the indications of the scenarios analysis are useful to enhance the environmental performance of these products. The downside is that the application of RothC requires additional data collection and expertise if compared to the execution of PEF studies.

**Keywords:** Life Cycle Assessment (LCA); product environmental footprint (PEF); carbon sequestration; cover crops; soil organic matter

## 1. Introduction

Soil organic carbon (SOC) is the carbon that remains in the soil after partial decomposition of any material produced by living organisms. It constitutes a key element of the global carbon cycle through atmosphere, vegetation, soil, rivers, and oceans [1]. Soils are the largest terrestrial organic carbon reservoir. In fact, soils contain a stock of carbon to a depth of 1 m, which is about twice as large as that in the atmosphere and about three times that in vegetation [2].

SOC is the main component of soil organic matter (SOM), which in turn supports key soil functions, since it is critical for the stabilization of soil structure and retention

and release of plant nutrients, and it allows water infiltration and storage. It is therefore essential to ensure soil health, fertility, and food production [1]. The loss of SOC indicates a certain degree of soil degradation. Unsustainable management practices such as excessive irrigation or leaving the soil bare endanger these soils, causing SOC loss and massive erosion [1]. On the other hand, observations from field experiments suggest that agricultural practices enhancing SOC level can also improve physical soil quality, reduce susceptibility to erosion, and outperform conventional systems as regards agricultural yields and yield stability, especially under drought stress [3,4].

SOC has also a crucial and obvious role to play in the global response to climate change [4]. In fact, the loss of SOC affects soil health and food production negatively and exacerbates climate change. When SOM is decomposed, greenhouse gases (GHG) are emitted to the atmosphere, contributing to warm up the planet. On the other hand, many soils have the potential to increase their SOC stocks, thus mitigating climate change by reducing the atmospheric $CO_2$ concentration [1]. In the "4 per 1000" initiative, a multi stakeholder platform aiming at increasing SOC storage through sustainable practices, it was calculated that a global increase of 0.4% of SOC in the top of agricultural soil (within 1 m) could offset 20–35% of global anthropogenic greenhouse gas emissions [5,6].

Both agricultural practices and land use change can modify considerably the gain, or loss, of SOC. As an example, Erb et al. [7] explored land use-induced alterations of biomass turnover and discovered that in the global average, biomass turnover is 1.9 times faster with land use in comparison to a hypothetical natural vegetation state. This acceleration affects all biomes roughly equally, but with large differences between land use types. For example, land conversion from forests to agricultural fields is responsible for 59% of the acceleration, and the use of forests and natural grazing land accounts for 26% and 15%, respectively. The authors conclude that land use significantly and systematically affects the fundamental trade-off between carbon turnover and carbon stocks.

Within the past several decades, Life Cycle Assessment (LCA) has been applied widely in examinations of the environmental burden of agricultural products and practices, but impacts on soil healthiness have often been neglected [8]. The increasing awareness on the importance of the environmental consequences of land use (and carbon-based GHG related emissions) has led to the need to include them in the recommended impact categories of LCA studies [9–12]. SOC, which is considered one of the most significant indicators of soil quality, has been proposed by some authors as an indicator in LCA of the impact of land use on its long-term ability to produce biomass [13–16]. The need to predict the organic carbon in soils has led—with increasing frequency in LCA studies—to the application of models able to evaluate its temporal trend. Goglio et al. [17] classified the models to account for SOC in agricultural LCAs into two groups: simple C models, and dynamic crop–climate–soil models. The first are based on a set of simple equations. Most of them consider the temporal dynamics of SOC but do not simulate crop production; therefore, C inputs are necessary to run the models. They listed the following among simple C models: ICBM (Introductory Carbon Balance Model) [18], C-TOOL [19], and RothC [20]. In the second group of models, they recorded DNDC (DeNitrification DeComposition) [21], DAYCENT, which is the daily time-step version of the CENTURY biogeochemical model [22], and CENTURY [23]. Among the several models available and reviewed in Goglio et al. [17], RothC has been widely applied in agriculture LCA studies since it first appeared in 2009. Hillier et al. [24] integrated SOC balance through RothC in life cycle analyses of energy crops. Cherubini and Ulgiati [25] analyzed the variation of SOC due to corn stover removal. Nguyen et al. [26] carried out a consequential LCA on dairy systems and adopted the RothC model in order to understand the effects of different agriculture approaches on SOC management. Yao et al. [27] combined the RothC model with LCA and demonstrated that the use of soybean instead of summer fallow can reduce persistently GHG emissions of wheat production. They stated that the RothC model coupled with LCA is an alternative method to predict the long-term impact of different cropping systems on GHG emissions. Morais et al. [28] applied the RothC model in order to calculate SOC change as an impact

indicator of land use in LCA. Boone et al. [16] used the RothC model to simulate the SOC evolution due to farm management and proposed SOC as indicator of environmental sustainability of agricultural systems. Lefebvre et al. [29] combined the RothC model and LCA in order to calculate the reduction of GHG emissions due to the application of biochar from sugarcane residues.

This paper describes the application of the RothC model to cultivation of an olive grove. The study was carried out in the framework of LIFE EFFIGE [30], a project that aims to enable Italian companies to measure their environmental footprint by adopting the Product Environmental Footprint (PEF) method [31]. This method is based on the life-cycle principles and aims to increase robustness, consistency, comparability, and reproducibility of LCA studies and to encourage communication of environmental performance of products. To this purpose, the method provides general guidance on how to develop specific product requirements (Product Environmental Footprint Category Rules—PEFCRs) for conducting life cycle impact assessment studies. A number of PEFCRs has been developed during the pilot phase of the European Commission PEF program [32], which lasted from 2015 to 2018, and numerous pilots are under development in the current transition phase. Several products have been considered, and large participation of the agri-food sector (olive oil, dairy products, feed, beer, pasta, etc.) has been recorded. In addition to indicating the most relevant impact categories for the product category, a PEFCR document specifies the so-called "additional environmental information" that must be included, and reported separately, in the PEF study. The impacts of land use change on biodiversity and indicators of SOC storage are examples of "additional information".

This study has three general objectives: (1) to evaluate the relevance of SOC storage, in terms of $CO_{2eq}$, if compared to total GHG emissions in the olive oil life cycle; (2) to test the applicability of the RothC model in the context of PEF studies; and (3) to understand how the results obtained by the RothC model can support the agricultural holdings and complement the information provided by a PEF study. To the best of our knowledge, no scientific papers have yet been published that test and discuss the use of RothC to complement the environmental footprint of olive oil.

## 2. Materials and Methods

### 2.1. RothC Model

RothC was initially developed to evaluate the carbon turnover in arable soils of the Rothamsted Long Term Field Experiments, but then its applicability has been extended to other ecosystems [33,34]. The RothC model simulates the turnover of organic carbon in non-waterlogged surface soils as a function of soil type, temperature, moisture content, and vegetation cover. It uses a monthly time step to calculate SOC (t C ha$^{-1}$) [20] through the knowledge of climatic and pedologic variables, land use, and soil management.

Stock and variation of SOC are calculated by mathematical equations that describe the physical and chemical processes involved. RothC splits the SOC into four active pools (i.e., Decomposable Plant Material—DPM; Resistant Plant Material—RPM; Microbial Biomass—BIO; Humified Organic Matter—HUM) and a small pool of Inert Organic Matter (IOM) not involved in turnover processes. In case of plants, the incoming carbon is split between DPM and RPM depending on the DPM/RPM ratio of the particular incoming plant material (from 0.25 for woodland to 1.44 for improved grassland). In the soil, DPM and RPM carbon pools decompose into BIO and HUM, while a portion is lost as $CO_2$. The partition between $CO_2$ and BIO+HUM (46% BIO and 54% HUM) is determined by the clay content of the soil. BIO and HUM, in turn, further decompose into $CO_2$, BIO, and HUM [35].

The following parameters calculated on a monthly basis are the inputs of the model: precipitation and potential evapotranspiration (mm); average temperature (°C); degree of soil cover (bare or vegetated); carbon inputs from crop residues (t C ha$^{-1}$) with the related DPM/RPM ratio; and carbon inputs from manure (t C ha$^{-1}$). Moreover, the following soil parameters are required: clay concentration (%); soil tillage depth (cm), and IOM content (t C ha$^{-1}$) calculated from the initial measured SOC value.

The outputs provided by RothC are the values of SOC and the four active pools that compose it, in addition to the carbon emitted as $CO_2$. The output time step can be monthly or annual. In this way, the distribution of carbon pools and their variation over time can be evaluated within simulations varying from years to centuries. The RothC model is available at the Rothamsted website [36].

### 2.2. Case Study

As mentioned in the introduction, the study area is cultivated with organic olive groves and is located in Lazio region (Central Italy). The soil map of Lazio Region [37] reports, for the study area, a pedologic environment characterized by slopes on prevalent sands that are locally uneven; the elevation ranges from 20 to 500 m a.s.l., and slope ranges from moderate to steep (6–35%). The soils in the area are classified as Calcaric Cambisols according to the Word Reference Base system [38] and have very large useful depth; good drainage; clayey loam texture; few coarse fragments in surface horizons; and common in underlying horizons. Soils also develop on a very calcareous substrate with weakly alkaline reaction on surface horizons and moderately alkaline in the underlying horizons [39]. The available analyses of surface soil samples confirm the texture type, with sand = 41%, silt = 26%, and clay = 33%. The reaction shows basically neutral pH values and a medium content of total organic carbon (TOC) with a concentration of 1.33% [40].

According to data referring to 2016 and 2017 provided by the meteorological website Arsial [41], the area has average monthly temperatures ranging from 7 °C (January) to 26 °C (August) and annual total rainfall of 723 mm. According to Köppen's climate classification as revised by Pinna [42], the area presents a "Csa" climate type, i.e., temperate with at least one "dry summer month", having rainfall <30 mm and average temperature >22 °C.

The case study is representative of the agricultural practices and production yields of the area. Data of agricultural working activities carried out in 2016 and 2017 were collected, including the presence of cover crops. The soil was bare for a period of approximately 5 months after harrowing (i.e., from June to October), and no manure was used. Only once was pelleted organic fertilizer, containing 35% of organic carbon, provided (0.25 t C ha$^{-1}$).

### 2.3. Scenarios and Implementation of Input Data

Four scenarios were simulated for the case study. The input data are summarized in Table 1, and the model was applied using a combination of site-specific and literature data. Each scenario considered different agricultural practices for the olive grove, while pedological and climate data were the same for all four scenarios. Input variable data (agricultural practices and climate) were repeated yearly, while pedological data were constant.

The first scenario (Scenario 1) took into account the current agricultural management, with olive tree pruning residues and cover crops, represented by mixed grassland. In the second scenario (Scenario 2), the replacement of mixed grassland with field beans (*Vicia faba minor*, L. 1753) was assumed in order to evaluate how this different agricultural practice affects soil carbon storage. In Scenarios 3 and 4, a contribution of pelleted organic fertilizer (0.25 t C ha$^{-1}$) was assumed every year in addition to the agricultural practices of Scenarios 1 and 2. In all scenarios, the depth of tillage was 15 cm, which was the maximum depth achieved by the harrowing interventions carried out on the soil.

Monthly average temperature and monthly precipitation were provided considering the years 2016–2017. In absence of primary data, the monthly potential evapotranspiration was obtained from the Müller database guide [43].

**Table 1.** Summary of the input data for the 4 scenarios. Agricultural practices are distinguished for each scenario, while pedological and climate data are the same. Numbers in brackets indicate the data source.

| | Scenario 1 | | | | Scenario 2 | | | |
|---|---|---|---|---|---|---|---|---|
| | Agricultural Practices | | | | Agricultural Practices | | | |
| **Month** | C Input | | | Soil Cover (1) | C input | | | Soil Cover (1) |
| | Cultural Residues (1) | DPM/RPM Ratio (1) | Fertilizer (2) | | Cultural Residues (1) | DPM/RPM Ratio (1) | Fertilizer (2) | |
| Jan | Pruning | 1.02 | - | Vegetated | Pruning | 1.30 | - | Vegetated |
| Feb | Pruning | 1.02 | - | Vegetated | Pruning | 1.30 | - | Vegetated |
| Mar | Pruning | 1.02 | - | Vegetated | Pruning | 1.30 | - | Vegetated |
| Apr | - | - | - | Vegetated | - | - | - | Vegetated |
| May | Mixed grassland | 1.02 | - | Vegetated | - | - | - | Vegetated |
| Jun | - | - | - | Bare | Field beans | 1.30 | - | Vegetated |
| Jul | - | - | - | Bare | - | - | - | Bare |
| Aug | - | - | - | Bare | - | - | - | Bare |
| Sep | - | - | - | Bare | - | - | - | Bare |
| Oct | - | - | - | Bare | - | - | - | Bare |
| Nov | - | - | - | Vegetated | - | - | - | Vegetated |
| Dec | - | - | - | Vegetated | - | - | - | Vegetated |
| | **Scenario 3** | | | | **Scenario 4** | | | |
| | Agricultural Practices | | | | Agricultural Practices | | | |
| **Month** | C Input | | | Soil Cover (1) | C Input | | | Soil Cover (1) |
| | Cultural Residues (1) | DPM/RPM Ratio (1) | Fertilizer (2) | | Cultural Residues (1) | DPM/RPM Ratio (1) | Fertilizer (2) | |
| Jan | Pruning | 1.02 | - | Vegetated | Pruning | 1.30 | - | Vegetated |
| Feb | Pruning | 1.02 | - | Vegetated | Pruning | 1.30 | - | Vegetated |
| Mar | Pruning | 1.02 | Organic fertilizer | Vegetated | Pruning | 1.30 | Organicfertilizer | Vegetated |
| Apr | - | - | - | Vegetated | - | - | - | Vegetated |
| May | Mixed grassland | 1.02 | - | Vegetated | - | - | - | Vegetated |
| Jun | - | - | - | Bare | Field beans | 1.30 | - | Vegetated |
| Jul | - | - | - | Bare | - | - | - | Bare |
| Aug | - | - | - | Bare | - | - | - | Bare |
| Sep | - | - | - | Bare | - | - | - | Bare |
| Oct | - | - | - | Bare | - | - | - | Bare |
| Nov | - | - | - | Vegetated | - | - | - | Vegetated |
| Dec | - | - | - | Vegetated | - | - | - | Vegetated |
| | Pedological and climatic parameters—same inputs for all scenarios | | | | | | | |
| | Clay in percentage (3); Soil depth (3); Initial measured SOC (4); Monthly precipitation in the years 2016–2017 (3); Monthly average temperature in the years 2016–2017 (3); Monthly potential evapotranspiration (1). | | | | | | | |
| | (1): Estimated [43] (2): Calculated from product label values | | | | (3): Measured data (4): Estimated from measured TOC by using [44] | | | |

Physical–chemical analyses of a representative soil sample from the study area were used for pedologic information. The initially measured SOC (29 t C ha$^{-1}$) was assessed on the basis of the percentages of sand, clay, and TOC, which were also used for estimating the bulk density with the Saxton and Rawls model [44]. Bare soil was assumed from the month following harrowing (Table 1).

The carbon contribution from olive pruning residues (0.50 t C ha$^{-1}$ y$^{-1}$) was calculated from an average value of fresh residues (2.2 t ha$^{-1}$ y$^{-1}$), considering a moisture content of 50% [45] and an organic carbon content of 45% on dry weight [46].

The entire cover crops are left in the soil after harrowing. To calculate carbon input, the dry matter input per hectare of the entire crop (whole crop residues) was estimated using the following equations [47]:

$$\text{Whole crop residues} = \text{GDW} + \text{AGR} + \text{BGR} \tag{1}$$

$$AGR = 0.325 \times GDW \tag{2}$$

$$BGR = 0.43 \, (AGR + GDW) \tag{3}$$

where GDW (grain dry weight) is the dry weight yield per hectare of crop mowing, AGR (above ground residues) is the above ground residue yield per hectare, and BGR (below ground residues) is the below ground residue yield per hectare. The coefficients in the expressions were those used in [47] for alfalfa. A yield of 18 t ha$^{-1}$ y$^{-1}$ on fresh weight [48] and a moisture content of 91% [33] were considered for estimating the GDW of mixed grassland (Scenarios 1 and 3).

On the other hand, the GDW of field beans (Scenarios 2 and 4) was estimated as being 4.5 t ha$^{-1}$ y$^{-1}$ of dry residues, adapting the indications given in [49,50] to the study site context. GDW, AGR, and BGR values were then multiplied by an organic carbon content of 45% [46]. The carbon inputs from mixed grassland and from field beans were then estimated equal to 0.9 and 3.8 t C ha$^{-1}$ y$^{-1}$, respectively.

Following the RothC handbook [35], a DPM/RPM ratio of 0.25 was assigned to pruning residues (biomass richer in lignin), and a value of 1.44 to cover crops (mixed grassland and field beans, biomass richer in cellulose). Since the software needed a single value of the DPM/RPM ratio, a weighted average value was calculated based on the different organic carbon inputs: the value was 1.02 for Scenarios 1 and 3, and 1.30 for Scenarios 2 and 4.

In all scenarios (Table 1), carbon inputs from olive pruning residues were assumed in January, February, and March, while those from cover crops were considered in May, at the mixed grassland harrowing time (Scenarios 1 and 3), and in June, at the field beans burying (Scenarios 2 and 4). In Scenarios 3 and 4, organic fertilizer was assumed to be spread in March.

In agreement with [33,35,51], the initial measured SOC and its constituent pools were assumed to be the result of a prior equilibrium condition. Firstly, the carbon inputs needed to match the initial measured SOC content were calculated by running the model in "inverse mode" and using the DPM/RPM ratio = 1.02.

Subsequently, the calculated inputs were used to estimate the pool initial values by running the model at steady-state (equilibrium mode). As a second step, the model was run in "forward mode" with the data from four scenarios. A 1000-year simulation was chosen to observe how long it takes the SOC levels of the four scenarios to reach the equilibrium.

### 2.4. How to Seize the Effect of SOC Storage on GHGS Emissions

The PEF study carried out on the olive oil produced in the study area is not available for publication, due to confidentialities issues. In order to discuss the relevance of soil carbon storage in comparison with the total GHG emissions in the olive oil life cycle, we decided to consider the results of the study by Iraldo et al. [52]. The choice of taking as a reference the value of Global Warming Potential (GWP100) calculated by Iraldo et al. is considered acceptable for the purpose of this paper, because the geographic areas investigated in the two study areas are similar, and olive production per hectare and olive oil yield per kg of olive are almost the same. Table 2 summarizes the main considered parameters.

**Table 2.** Main parameters characterizing the study by Iraldo et al. [52].

| Olive Production | Oil Production | Oil Yield | Functional | GWP100 |
|---|---|---|---|---|
| kg ha$^{-1}$ y$^{-1}$ | kg ha$^{-1}$ y$^{-1}$ | % | unit (FU) | kg CO$_{2eq}$ FU$^{-1}$ |
| 2.733 | 419.4 | 15 | 1 kg of extra virgin olive oil | 3.63 |

The relevance of soil carbon storage in terms of CO$_2$ equivalent was evaluated in a time-frame of 100 years in order to align it with the time-frame of the impact category GWP100 calculated by Iraldo et al. [52]. The time-frame is the same as for the impact category GWP recommended by the PEF method [31]. In order to compare the GWP100

results reported in [52] to RothC results, the GWP100 value of 1 kg of olive oil was multiplied by the amount of oil produced by one hectare per year, thus obtaining the value of 1.522 kg $CO_{2eq}$ ha$^{-1}$ y$^{-1}$.

The annual *SOC* variation in a period of 100 years was calculated for the four scenarios according to the following formula (Equation (4)) [27]:

$$\Delta SOC \left( \text{t C ha}^{-1}\text{y}^{-1} \right) = \frac{SOC_{final}\left( \text{t C ha}^{-1} \right) - SOC_{initial}\left( \text{t C ha}^{-1} \right)}{T\left( y \right)} \tag{4}$$

where $T(y)$ is the time period between the initial and final *SOC* value (i.e., 100 years), $SOC_{final}$ is the *SOC* value at $T(y)$, and $SOC_{initial}$ is 29 t C ha$^{-1}$ (see Section 2.3).

Finally, in order to evaluate how much the annual *SOC* variation ($\Delta SOC$) affects the GWP100 of the olive oil yearly produced by one hectare, the following equation (Equation (5)) was used:

$$\Delta CO_{2eq} \left( \text{kg CO}_{2eq}\text{ha}^{-1}\text{y}^{-1} \right) = \Delta SOC \left( \text{t C ha}^{-1}\text{y}^{-1} \right) \cdot \frac{44}{12} \cdot 1000 \left( \text{kg t}^{-1} \right) \tag{5}$$

where 44 and 12 are the $CO_2$ and C molecular and atomic weights, respectively.

## 3. Results and Discussion

### 3.1. Analysis of Scenarios Simulated by RothC

The active pools, estimated by RothC, in the studied site at the beginning (year 1) were DPM = 0 t C ha$^{-1}$, RPM = 3.8 t C ha$^{-1}$, BIO = 0.5 t C ha$^{-1}$, and HUM = 22.2 t C ha$^{-1}$, while the IOM was 2.3 t C ha$^{-1}$.

Figures 1–4 show the results, for scenarios 1–4, respectively, of carbon pool trends obtained with RothC, using a 1000-year simulation, with data output at the end of each year. In Scenario 1, DPM pool goes to zero before the end of each year, while in Scenario 2, a small constant stock remains ($5.2 \times 10^{-3}$ t C ha$^{-1}$), due to the contribution of the field beans. Null or very low values of this pool can be explained by its fast turnover. Scenario 2 shows an increase in all other pools (HUM, RPM, and BIO), although at varying degrees and different settling times to equilibrium. This results in an increase of SOC, which reaches 45 t C ha$^{-1}$ after 745 years. On the other hand, Scenario 1, corresponding to the current agricultural management of the olive grove, shows a significant decrease in SOC, which passes from 29 to 11 t C ha$^{-1}$ after 447 years.

This suggests that carbon inputs provided by plant residues from current farm management are not adequate to maintain the current SOC in the long term. On the other hand, the use of field beans could lead to a significant increase in the soil carbon stock.

Scenario 3 (Figure 3) has a pattern similar to Scenario 1. Carbon input by organic fertilizer, in addition to pruning and mixed grassland residues, leads to a slightly less pronounced decrease of carbon pools. Soil organic carbon stabilizes at 12 t C ha$^{-1}$ after 451 years. However, it is evident that the organic fertilizer provided is not sufficient to maintain the carbon stock equal to initial equilibrium conditions.

Scenario 4 (Figure 4) has a similar trend to Scenario 2. The addition of organic fertilizer is reflected in a slight increase in carbon pools and SOC compared to Scenario 2. Soil organic carbon takes longer to reach stable values (47 t C ha$^{-1}$ after 907 years); this might be partly due to the humic component (HUM) directly supplied to the soil via organic fertilizer, which has a much longer turnover time than DPM and RPM.

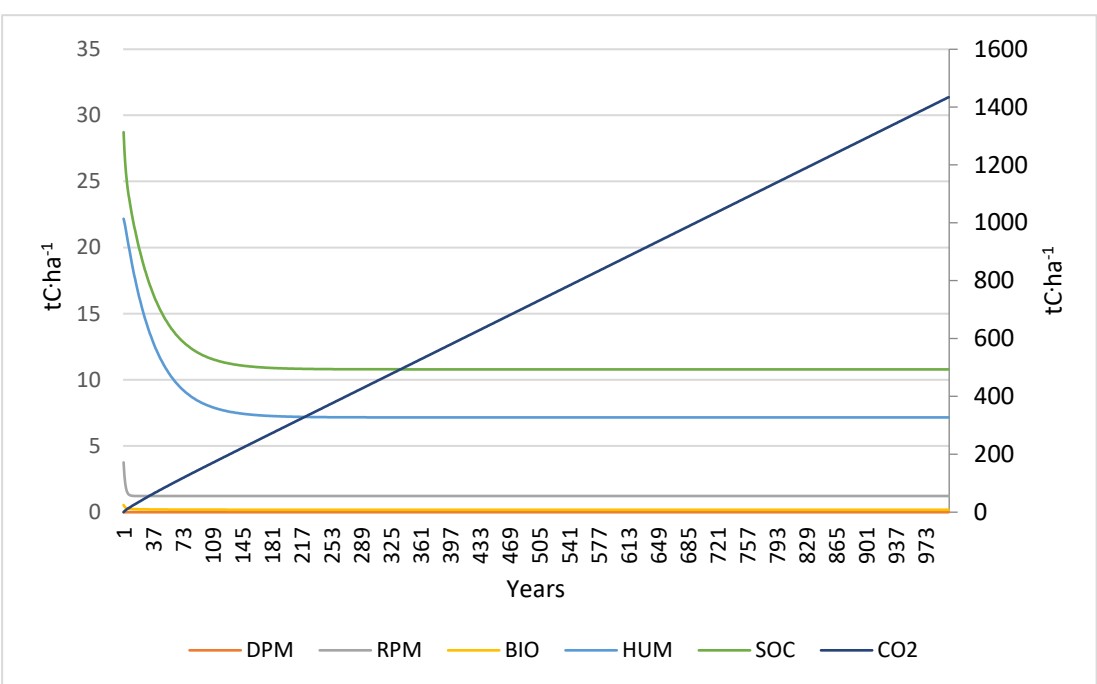

**Figure 1.** Scenario 1: stocks of soil carbon pools (**left ordinate**) and cumulated carbon as $CO_2$ (**right ordinate**). Values are shown at the end of each year for 1000 years.

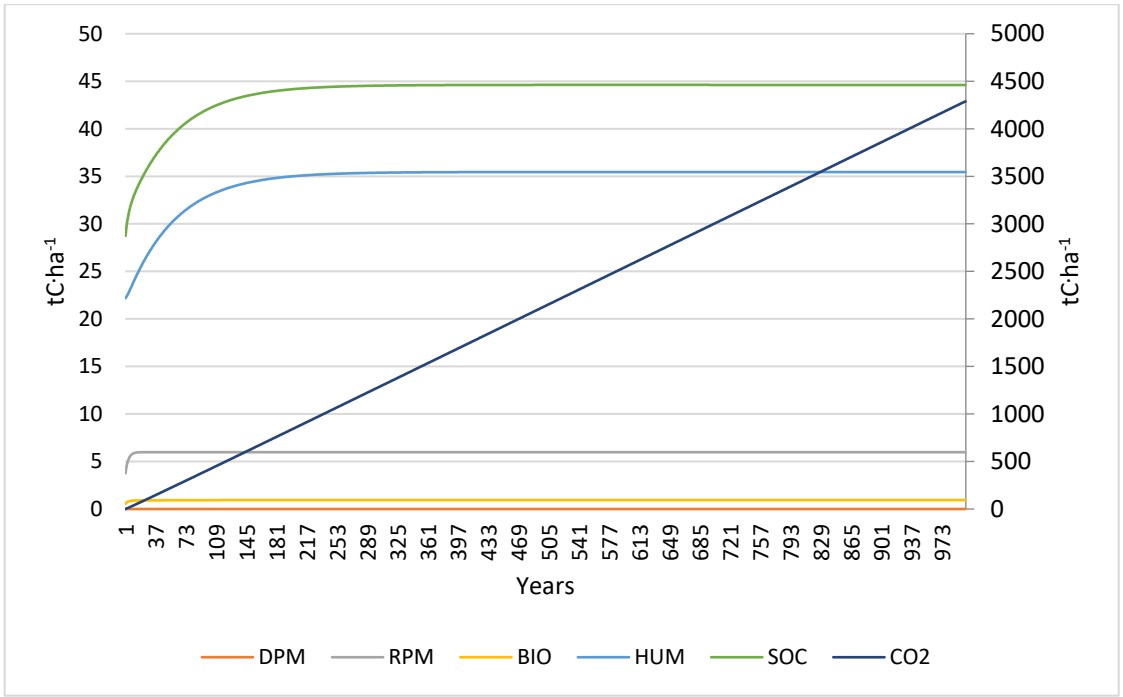

**Figure 2.** Scenario 2: stocks of soil carbon pools (**left ordinate**) and cumulated carbon as $CO_2$ (**right ordinate**). Values are shown at the end of each year for 1000 years.

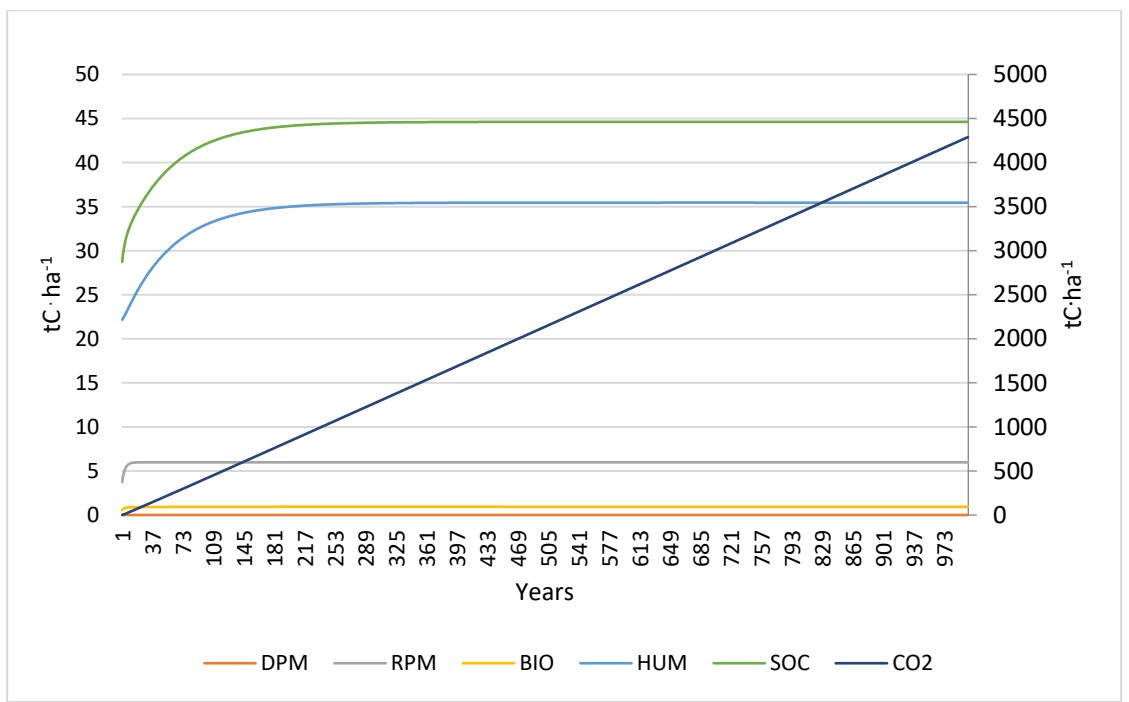

**Figure 3.** Scenario 3: stocks of soil carbon pools (**left ordinate**) and cumulated carbon as $CO_2$ (**right ordinate**). Values are shown at the end of each year for 1000 years.

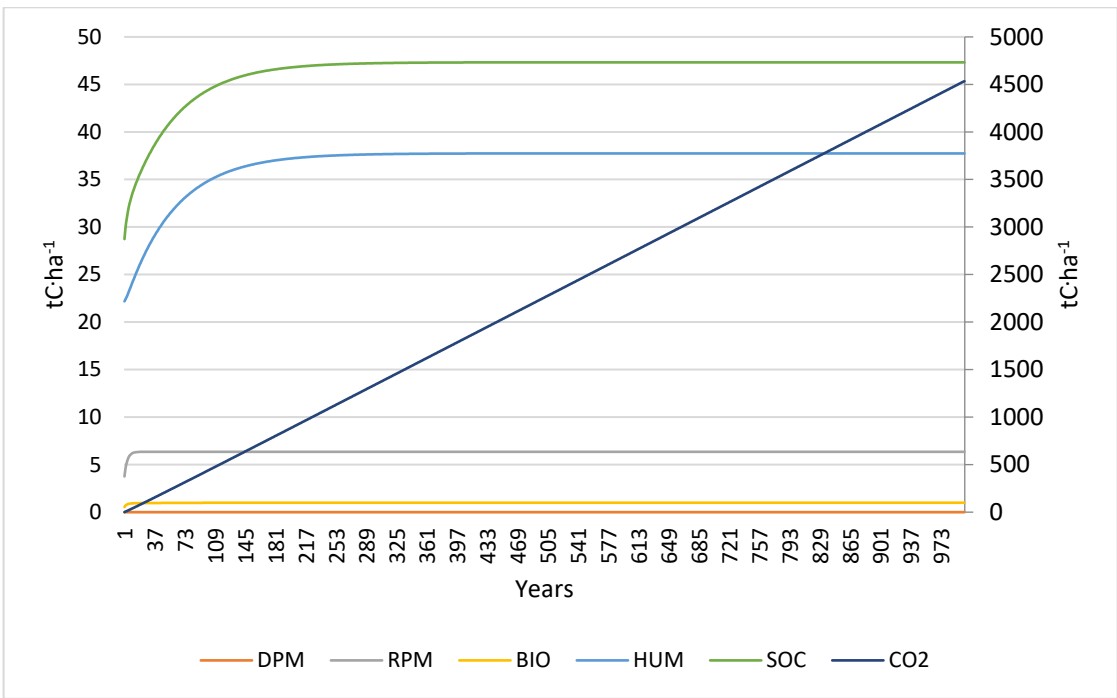

**Figure 4.** Scenario 4: stocks of soil carbon pools (**left ordinate**) and cumulated carbon as $CO_2$ (**right ordinate**). Values are shown at the end of each year for 1000 years.

In general, however, the amount of organic fertilizer provided in Scenarios 3 and 4 is not enough to observe consistent changes in carbon pools, if compared to Scenarios 1 and 2, respectively.

For the whole simulation period, the cumulated carbon inputs to the soil as plant residues and organic fertilizer, and the cumulated carbon output lost as $CO_2$, were com-

pared. As Table 3 shows, the net balance values, input–output, correspond to the SOC variation that occurred in the four scenarios during the 1000-year simulation. In Scenarios 1 and 3 where SOC decreases, the net balance is negative, while where SOC increases (Scenarios 2 and 4), the net balance is positive.

**Table 3.** Net balance between cumulated carbon in inputs and outputs after 1000 years of simulation.

|  | Input (Residues + FYM) | Output ($CO_2$) | Input−Output |
|---|---|---|---|
|  | t C ha$^{-1}$ | t C ha$^{-1}$ | t C ha$^{-1}$ |
| Scenario 1 | 1416 | 1434 | −18 |
| Scenario 2 | 4305 | 4289 | 16 |
| Scenario 3 | 1663 | 1680 | −17 |
| Scenario 4 | 4552 | 4534 | 18 |

### 3.2. Relevance of Soil Carbon Storage in Terms of $CO_2$ Equivalent

Table 4 shows the results of SOC obtained by RothC with a simulation of 100 years for the four scenarios, the annual variation of SOC in a period of 100 years and in an area of 1 hectare (ΔSOC) obtained by Equation (4), and the corresponding annual $CO_{2eq}$ variation obtained by Equation (5) (see Section 2.4).

**Table 4.** SOC values obtained by RothC refer to a period of 100 years, related annual variation of SOC (ΔSOC), and annual variation of $CO_{2eq}$ (ΔCO$_{2eq}$) for each scenario.

| Scenario | SOC (at 100 Years) t C ha$^{-1}$ | ΔSOC t C ha$^{-1}$ y$^{-1}$ | ΔCO$_{2eq}$ t CO$_{2eq}$ ha$^{-1}$ y$^{-1}$ |
|---|---|---|---|
| Scenario 1 | 12 | −0.17 | −0.62 |
| Scenario 2 | 42 | 0.13 | 0.48 |
| Scenario 3 | 13 | −0.16 | −0.59 |
| Scenario 4 | 44 | 0.15 | 0.55 |

As Table 4 shows, Scenarios 1 and 3 have negative ΔSOC and therefore negative ΔCO$_{2eq}$, whereas Scenarios 2 and 4, where cover crops are included in the agricultural management of the olive grove, have positive values. These outcomes mean that in Scenarios 1 and 3 there is a net $CO_2$ emission; instead in Scenarios 2 and 4, there is a net carbon storage in soil corresponding to savings of 0.48 and 0.55 t of $CO_2$ emission per year per hectare, respectively.

Table 5 compares the total GHG emissions in the olive oil life cycle [52] to the annual variation of $CO_{2eq}$ stored by soil in a period of time of 100 years (ΔCO$_{2eq}$) calculated by Equation (5) from the RothC model output data for the four scenarios.

**Table 5.** GHG emissions of olive oil life cycle by Iraldo et al. [52] and $CO_{2eq}$ storage (positive values) or release (negative values) calculated using RothC model (both expressed as kg CO$_{2eq}$ ha$^{-1}$ y$^{-1}$).

| Scenario | GWP100 | ΔCO$_{2eq}$ Stored by Soil |
|---|---|---|
| Scenario 1 | 1522 | −620 |
| Scenario 2 | 1522 | 480 |
| Scenario 3 | 1522 | −590 |
| Scenario 4 | 1522 | 550 |

The comparison shows that Scenarios 2 and 4 have positive values of ΔCO$_{2eq}$ stored in the soil because there is an annual net carbon storage, due to the field beans cover crop.

On the contrary, for the other scenarios where the $\Delta CO_{2eq}$ stored in the soil is negative, there are annual net emissions of $CO_2$ also from the soil, which add emissions along the supply chain. Moreover, the magnitude of the results highlights that the soil carbon storage is relevant when compared to the total GWP values of the olive oil life cycle. Furthermore, the results show that different agricultural practices (e.g., sowing of cover crops) strongly affect carbon storage or depletion.

Finally, it should be emphasized that organic farming is not necessarily synonymous with "protection of the carbon present in the soil". In fact, if carbon inputs are insufficient to maintain the carbon balance in the soil, the soil is depleted [53,54]. However, some agronomic practices typical of organic management (e.g., cover crops, minimum tillage, and organic fertilization) generally lead to a lower risk of SOC depletion, in comparison to conventional management. This effect has been observed in most experimental studies carried out in different soil contexts [55].

### 3.3. Applicability of the RothC Model in LCA and PEF Studies

In order to evaluate the applicability of the model for its use in LCA and PEF studies, in particular for companies to which PEF method is addressed, three features were considered: availability of input data, user-friendliness of the software, and ease of results interpretation. The required data can be divided into four groups: climatic, pedologic, agricultural management, and carbon-input information. Generally, temperature and precipitation are easily available on specialized websites (weather and climate information services). Obviously, the possibility to find site-specific data depends on the density of the network. Finding evapotranspiration data may be rather difficult since solar radiation data are needed, and usually this parameter is scarcely monitored by the weather services. Nevertheless, radiation can be obtained also by technical literature suggested by the RothC user's guide [35] or by large-scale weather information systems [56]. As regards, pedologic data, due to the soil variability, ad hoc chemical–physical analysis should be used, rather than technical literature. In any case, only three values are required for each soil sample: percentage of clay, bulk density, and total organic carbon. These are quite easy and inexpensive lab analyses. The carbon input assessment of crop residues, due to lack of direct measurements for the above and below ground plant portions, must be estimated empirically. To this purpose, numerous regression equations have been proposed for the most common crops [47,57]. However, in our opinion, estimating carbon inputs can be more difficult and less reliable for marginal crops or for crops growing under particular conditions (e.g., drought, nutrient-poor soil). The case study shows that in most cases, the amount and type of data necessary to implement the RothC model are not expansive and time-consuming. However, the availability of good quality input data depends also on the farm site-specific characteristics and on the chosen crop plants.

Regarding user-friendliness, the RothC user's guide is a precious help because it provides detailed indications on the entry interfaces. The last step, i.e., results interpretation, although supported by graphs, appears somehow difficult. The user's guide supplies information to understand algorithms and final values but specific expertise is needed to interpret and explain results from pedologic and agronomic viewpoint. In conclusion, the RothC model appears easy to use, but the user should be supported by a soil expert to make a correct interpretation of the results that otherwise could not be adequately exploited.

### 3.4. Suggestions from the RothC Model to Improve the Farm Environmental Performances

The RothC model results highlight that adherence to the organic agricultural practices does not guarantee by itself that soil quality, in terms of organic carbon content, will be preserved. The cover crops choice (e.g., choosing field beans over grassland) is one of the main critical factors. Nevertheless, Hábová et al. [53] stated that in spite of less carbon input, organic farming was more stable in comparison to intensive farming. The intensive farming system was much more affected by climatic condition and plant residues input. Another important issue is the organic fertilization, which should be increased to enhance

the organic carbon stock in the soil, especially if manure is available from farms nearby. Using the Scenario 1 parameters as a reference, it was estimated that the amount of carbon to be contributed as farmyard manure, to maintain the initial SOC value (29 t C ha$^{-1}$) for at least 100 years, would be approximately 2.5 t C ha$^{-1}$ y$^{-1}$. In order to reduce external inputs, the use of all other residues of the olive oil production chain (i.e., virgin, exhausted or stoned pomace, and olive mill wastewater) should be considered as carbon inputs for the soil, when applicable [58]. The contribution of C from field beans should be verified by field direct measurements of biomass samples per unit area to obtain more accurate results. Similarly, the actual increase in SOC should be monitored annually. If the measured data do not agree with the model values, the input data could be modified to tune the simulation to the real data. In case the field beans do not provide the expected stocks of C, and when the C deficit cannot be compensated by the use of manure or residues from the olive industry, it would be advisable to choose other legumes, e.g., *Trifolium hirtum* or *Medicago littoralis*, which are indicated as less water-demanding and more resistant to drought [59].

## 4. Conclusions

The RothC model was applied to calculate soil carbon storage as "additional environmental information" in the framework of a wider activity aimed at producing the PEF of an Italian organic extra virgin olive oil. The model supports the evaluation of the contribution of cultivation practices to GHG emissions mitigation, allows for comparison of different scenarios, and can provide suggestions to improve the environmental performance of farms. Regarding the first research question, it is possible to conclude that calculation of SOC storage is very relevant in the life cycle of olive oil. In all analyzed scenarios, the results obtained over a 100-year time horizon show that the contribution, in terms of CO$_{2eq}$ associated with the organic carbon stored in the soil, is remarkable compared to the total GHG emissions of the olive oil life cycle. Therefore, including the evaluation of soil carbon storage in PEF studies of olive oil is important for a comprehensive assessment of their environmental performance.

The application of the RothC model to the case study has allowed the second research question to be answered. The applicability of the RothC model in the context of PEF studies appears to be an essential but complex task. A reliable application of the RothC model requires collecting additional data, if compared with data collection for the PEF study. These data are site specific, often are not under farmers' control, and additional expertise is needed for their correct interpretation. All this work adds to the time- and resource-consuming activity for data collection and elaboration for the PEF studies, which may be a barrier to the widespread use of the PEF method. In our opinion, even if carbon storage is a significant element that adds information to the impact categories included in the PEF method, consensus needs to be reached about its integration in PEF, and much work must be done to increase practicability and to address its use for increasing sustainability of agricultural practices.

Finally, the study has shown that the use of the RothC model can support agricultural holdings and complement the information provided by a PEF study. The cultivation practices and the use of specific types of cover crops have been confirmed to be a discriminating factor for mitigating or increasing life cycle emissions. RothC can enrich PEF results and support the design of different scenarios and reduction strategies to decrease GHG emissions.

**Author Contributions:** Conceptualization, V.F., P.B., A.B., E.N., G.Z. and S.R.; methodology, V.F., D.Z., A.B. and S.R.; software, V.F., D.Z. and C.S.; validation, P.B., A.B. and S.R.; formal analysis, P.B. and A.B.; investigation, D.Z., C.S., V.F., P.B., G.Z. and E.N.; resources, P.B., V.F. and S.R.; data curation, D.Z., C.S., P.B. and V.F.; writing—original draft preparation, V.F., P.B., D.Z., C.S., A.B. and S.R.; writing—review and editing, V.F., P.B., D.Z., C.S., A.B. and S.R.; visualization, C.S. and D.Z.; supervision, S.R.; project administration, P.B.; funding acquisition, P.B., V.F. and S.R. All authors have read and agreed to the published version of the manuscript.

**Funding:** This activity is part of the project LIFE EFFIGE (Environmental Footprint for Improving and Growing Eco-efficiency, LIFE16 ENV/IT/000172), co-financed by the LIFE Programme of the European Union.

**Data Availability Statement:** Data sets are available upon request to the authors.

**Conflicts of Interest:** The authors declare no conflict of interest. The funders had no role in the design of the study; in the collection, analyses, or interpretation of data; in the writing of the manuscript; or in the decision to publish the results.

### List of Acronyms

AGR: Above Ground Residues

BGR: Below Ground Residues

BIO: Microbial Biomass

DPM: Decomposable Plant Material

FU: Functional Unit

GDV: Grain Dry Weight

GHG: Green House Gasses

GWP100: Global Warming Potential at 100 years

HUM: Humified Organic Matter

IOM: Inert Organic Matter

LCA: Life Cycle Assessment

LCI: Life Cycle Inventory

PEF: Product Environmental Footprint

PEFCRs: Product Environmental Footprint Category Rules

RPM: Resistant Plant Material

SOC: Soil Organic Carbon

SOM: Soil Organic Matter

TOC: Total Organic Carbon

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
