# Peer review of "The RothC Model to Complement Life Cycle Analyses: A Case Study of an Italian Olive Grove"

_sustainability, doi:10.3390/su14010569_

Round 1

Reviewer 1 Report

The reviewer appreciates the authors' challenges in applying the RothC model to an olive grove in Italy. The followings are the reviewer's comments for revisions. I (the reviewer) am not a specialist in LCA study field or an olive grower. Please reject my comments if the authors found any reviewer's misunderstanding in the following. 

1) The title could be revised according to the contents. Or the contents could be revised according to the title. Probably, the authors could more clearly demonstrate the answer to the question raised in the title by showing the difference between LCA results with SOC data using the RothC model and those without SOC data. If it is impossible or difficult, the reviewer recommends the authors will emphasize the difficulty and main focus of this study.  

2) Among the three objectives (L.122-128), the first one was responded in conclusion but the second and third one remain unclear because the results of PEF studies are not referred to. As pointed in 1) above, to discuss the objective 2 and 3, the authors need to include the SOC data from the RothC model in the PEF studies. 

3) "3. Results" contains discussion. 3.3 and 3.4 seem to be discussion. The reviewer suggests that "3. Results" is to be "3. Results and discussion" or to be divided into two parts: "3. Results" and "4. Discussion".

4) The period of analysis covers 1,000 years but the curves after 100 years are relatively static. Therefore, the reviewer recommends more focused and intensive discussion on coming 10, 30, 50..., and 100 years (as the authors demonstrated in Table 4 and 5). For example, long term management plans of replanting or regeneration of olive trees may affect the total carbon stock of the olive grove. How many years do farm managers consider their general management models of olive groves in the target area? 

5) Some minor points:

L.158: the URL of [36] can be deleted in the main text. 

L.250: In "SOC Storage on Climate Change", the relation between SOC storage and climate change is not clear. "SOC Storage on GHGs Emissions" might be more understandable. 

L.572: "Agricoltural education online." This English site name is not necessary? Or it could be "Instruzione Agraria online" as original. 

That is all. I am sorry of the late response. Thank you.  

Author Response

Comment 1) The title could be revised according to the contents. Or the contents could be revised according to the title. Probably, the authors could more clearly demonstrate the answer to the question raised in the title by showing the difference between LCA results with SOC data using the RothC model and those without SOC data. If it is impossible or difficult, the reviewer recommends the authors will emphasize the difficulty and main focus of this study.

Answer 1): thank you for the suggestion. In order to take up the suggestion, we prefer to change the title of the paper in “RothC Model to Complement Life Cycle Analyses: the case study of an Italian olive grove”. In this way, we highlight the supporting role of RothC (“complement”) to the life cycle studies (“life cycle analyses”).

Comment 2) Among the three objectives (L.122-128), the first one was responded in conclusion but the second and third one remain unclear because the results of PEF studies are not referred to. As pointed in 1) above, to discuss the objective 2 and 3, the authors need to include the SOC data from the RothC model in the PEF studies.

Answer 2): thank you for the suggestion. We have improved our paper by including these recommendations. We have added the major results of paragraph 3.3 “Applicability of RothC model in LCA and PEF studies” and 3.4 “Suggestions from RothC model to improve the farm environmental performances” in Chapter 4 “Conclusions”.

Comment 3) "3. Results" contains discussion. 3.3 and 3.4 seem to be discussion. The reviewer suggests that "3. Results" is to be "3. Results and discussion" or to be divided into two parts: "3. Results" and "4. Discussion".

Answer 3): thank you for the suggestion. We think that it is more effective maintain results and discussion together in the text, therefore we have changed the title of chapter 3 in “Results and discussion”.

Comment 4) The period of analysis covers 1,000 years but the curves after 100 years are relatively static. Therefore, the reviewer recommends more focused and intensive discussion on coming 10, 30, 50..., and 100 years (as the authors demonstrated in Table 4 and 5). For example, long term management plans of replanting or regeneration of olive trees may affect the total carbon stock of the olive grove. How many years do farm managers consider their general management models of olive groves in the target area?

Answer 4): Thank you for the suggestion. The reviewer is correct. The most significant changes in monitored parameters occur in the first 2-3 centuries. However, although smaller in magnitude, changes also occur later. For example, in scenario 2, stable SOC values are reached after 745 years, while in scenario 4, stable values are reached after 907 years. Certainly, we could have considered a shorter time period, but in order to understand the full performance of the treatments, we decided to extend the monitored period to 1000 years. Also because, an olive tree can live more than 1000 years.

Comment 5) Some minor points:

Answer 5): L.158: the URL has been deleted; L.250: we have changed the title of the paragraph in "SOC Storage on GHGs Emissions"; L.572: we have let the title of the site "Instruzione Agraria online" as original.

Reviewer 2 Report

The reviewing manuscript topic is to Is Soil Organic Carbon a Relevant Additional Information to Complement LCA Studies? The RothC Model Applied to an Olive Grove in Italy. The manuscript seems to be interesting but needs to be revised.
Would you please revise the title and change the question to your objective
Line 41, in fact, they?!! What are they? Would you please write clear
Line 55 sentence needs a verb
Line 105 – 121 please revise the English grammar 
The method and materials are clear. 
Line 282- 416 Please separate result and discussion
Need details about data availability  
References must be checked with Instructions for Authors. 
A list of abbreviations should also be added at the beginning of the manuscript.

Author Response

Comment: Would you please revise the title and change the question to your objective.

Answer: Thank you for the suggestion. We have changed the title of the paper with the aim of focusing on the role of support that RothC can have in the LCA studies. We think that now the title is more consistent with the aims of the paper.

Comment: Line 41, in fact, they?!! What are they? Would you please write clear

Answer: Thank you for the suggestion. We have changed “they” in “soils”

Comment: Line 55 sentence needs a verb

Answer: Thank you for the suggestion. We have changed the sentence; we hope that now it is more understandable.

Comment: Line 105 – 121 please revise the English grammar

Answer: Thank you for the suggestion. We have changed this part of the paper, we hope that now it is more understandable.

Comment:  The method and materials are clear.

Answer: thank you.

Comment: Line 282- 416 Please separate result and discussion

Answer: thank you for the comment. About chapter 3, the referee n° 1 considers both the possibilities 1) to maintain results and discussion together (and in the meantime to change the title of chapter 3 in “Results and discussion”) and 2) to separate result and discussion in two chapters. We would prefer to maintain the organisation of the paper that illustrates results and discussion in the same chapter. In our opinion, this organisation permits a greater understanding of the text and a greater effectiveness of the figures. Obviously, if the referee n° 2 thinks that it is unacceptable to have results and discussion in the same chapter, we are ready to separate them.

Comment: Need details about data availability

Answer: unfortunately, we do not understand the request of the referee. Since above he/she affirmed that “method and materials are clear”, we think that the phrase “Need details about data availability” refers to “Results and discussion” or to “Conclusions”. Paragraph 3.3 (Applicability of RothC model in LCA and PEF studies) discusses the availability of four groups of the required data (climatic, pedologic, agricultural management and input of carbon). In this paragraph we have added some further details about data availability.  We hope that these changes are in the desired direction.

Comment: References must be checked with Instructions for Authors.

Answer: Thank you for the suggestion. We have checked the list of references according to Instructions for Authors.

Comment: A list of abbreviations should also be added at the beginning of the manuscript.

Answer: thank you for the suggestion. We have added a list of abbreviation.

Round 2

Reviewer 1 Report

The manuscript has been revised according to the reviewer's comments. Thank you for the responses. The reviewer would also like to learn from the authors' further studies on both short term and long term options for mitigating GHGs in different agricultural systems.   

Reviewer 2 Report

Dear Authors

well done

I accept last version of article

best regards